DTDHM: detection of tandem duplications based on hybrid methods using next-generation sequencing data

Yuan Tianting 1
Dong Jinxin 1 dongjinxin@lcu-cs.com
Jia Baoxian 1
Jiang Hua 1 jianghua@lcu-cs.com
Zhao Zuyao 2
Zhou Mengjiao 1
1 School of Computer Science and Technology, Liaocheng University , Liaocheng , China
2 Orthopedics Department, Liaocheng People’s Hospital , Liaocheng , China
Galitsyna Aleksandra
Electronic publication date: 2024 Jul 26
Publication date: 2024
Volume: 12
Electronic Location ID: e17748
Received 2024 Jan 19; Accepted 2024 Jun 24
Copyright: © 2024 Yuan et al.
Copyright year: 2024
Copyright holder: Yuan et al.
License: This is an open access article distributed under the terms of the Creative Commons Attribution License, which permits unrestricted use, distribution, reproduction and adaptation in any medium and for any purpose provided that it is properly attributed. For attribution, the original author(s), title, publication source (PeerJ) and either DOI or URL of the article must be cited.
License URL: https://creativecommons.org/licenses/by/4.0/

Keywords: Tandem duplications, Structural variations, Hybrid methods, K-nearest neighbor algorithm, Next-generation sequencing, Short-read sequence

Funding: Discipline with Strong Characteristics of Liaocheng University–Intelligent Science and Technology 319462208 This work was supported by Discipline with Strong Characteristics of Liaocheng University–Intelligent Science and Technology under Grant 319462208. The funders had no role in study design, data collection and analysis, decision to publish, or preparation of the manuscript.

==============================
Background

Tandem duplication (TD) is a common and important type of structural variation in the human genome. TDs have been shown to play an essential role in many diseases, including cancer. However, it is difficult to accurately detect TDs due to the uneven distribution of reads and the inherent complexity of next-generation sequencing (NGS) data.

Methods

This article proposes a method called DTDHM (detection of tandem duplications based on hybrid methods), which utilizes NGS data to detect TDs in a single sample. DTDHM builds a pipeline that integrates read depth (RD), split read (SR), and paired-end mapping (PEM) signals. To solve the problem of uneven distribution of normal and abnormal samples, DTDHM uses the K-nearest neighbor (KNN) algorithm for multi-feature classification prediction. Then, the qualified split reads and discordant reads are extracted and analyzed to achieve accurate localization of variation sites. This article compares DTDHM with three other methods on 450 simulated datasets and five real datasets.

Results

In 450 simulated data samples, DTDHM consistently maintained the highest F1-score. The average F1-score of DTDHM, SVIM, TARDIS, and TIDDIT were 80.0%, 56.2%, 43.4%, and 67.1%, respectively. The F1-score of DTDHM had a small variation range and its detection effect was the most stable and 1.2 times that of the suboptimal method. Most of the boundary biases of DTDHM fluctuated around 20 bp, and its boundary deviation detection ability was better than TARDIS and TIDDIT. In real data experiments, five real sequencing samples (NA19238, NA19239, NA19240, HG00266, and NA12891) were used to test DTDHM. The results showed that DTDHM had the highest overlap density score (ODS) and F1-score of the four methods.

Conclusions

Compared with the other three methods, DTDHM achieved excellent results in terms of sensitivity, precision, F1-score, and boundary bias. These results indicate that DTDHM can be used as a reliable tool for detecting TDs from NGS data, especially in the case of low coverage depth and tumor purity samples.

Introduction

Genomic structural variation (SV) is one of the main types of genetic variations besides single nucleotide variation and slight nucleotide variation. SVs affect about 0.5% of the genome of a given individual (Eichler, 2012) and explain the characteristics of diversity in organisms and populations (Conrad et al., 2010). SVs include deletions, insertions, duplications, inversions, and translocations of DNA fragments (Balachandran & Beck, 2020). As sequencing costs have declined, the study of SVs has increased, making SVs a current hot spot in genomics research.

Tandem duplications (TDs) are regions of multiple adjacent sequence copies in genomic DNA, accounting for about 10% of the human genome (Valia & Zhang, 2006). TDs are highly variable between individuals and have high mutation rates in the genome due to replication errors that arise during cell division. The mutations in TDs can lead to expansion diseases, gene silencing, and rapid morphological variation. Studies have shown that TDs may be essential mechanisms of cancer gene activation. Xing et al. (2019) used whole-genome sequencing to reveal novel tandem duplication hotspots and prognostic mutational signatures in gastric cancer. Willis et al. (2017) proposed that BRCA1 is closely related to the formation of 10 kb TDs in ovarian cancer. Furthermore, Trost et al. (2020) suggested that the expansion of TD genes plays a vital role in the genetic etiology and phenotypic complexity of autism spectrum disorder. Therefore, TDs are the mechanisms of many human diseases (Ibañez et al., 2022; Newman et al., 2015). Accurately detecting TDs is necessary for identifying pathogenic genes and developing targeted drugs.

There are currently five strategies for detecting structural variations using next-generation sequencing (NGS): paired-end mapping (PEM), split read (SR), read depth (RD), de novo assembly (AS), and a combination of the these strategies (CB). These strategies are applied in different situations. PEM uses the salient features of discordant reads to infer the existence of SVs (Cleal & Baird, 2022; McLaughlin, 2021). However, it is unsuitable for the case where the insertion sequence is larger than the average insertion. It cannot detect fragment duplication in the SVs of the low-complexity region. SR splits and aligns the reads that cannot be completely matched to the reference genome. The detection resolution of SR can reach the base pair (bp) level, however, SR cannot reliably detect the insertion sequence of large fragments (Schroder et al., 2014). RD highlights duplication or deletion regions by studying differences in mapping depth and read distribution (Yuan et al., 2020, 2021; Zhang et al., 2022), but RD cannot accurately detect the boundaries of regions where SVs occur. AS constructs short DNA fragments into longer contiguous sequences through algorithms such as sequence alignment and sequence merging (Kavak et al., 2017; Zhuang & Weng, 2015). However, AS is computationally expensive and unsuitable for detecting long repetitive regions. CB integrates the above four strategies to highlight the advantages of each method, helping to improve the precision and sensitivity of test results (Eisfeldt et al., 2017; Soylev et al., 2019). Of these five methods, CB analyzes the changes of SVs the most effectively.

In recent years, many methods have been developed to detect TDs from NGS data. VNTRseek maps sequencing reads to a set of reference TDs. VNTRseek identifies putative TDs based on the difference between the copy number of the reference and its mapped reads (Gelfand et al., 2014), but it is only suitable for detecting microsatellite TDs. ScanITD performs a stepwise seeding and realignment procedure for ITD detection. It uses a string rotation method to determine whether the inserted sequence is a duplicate sequence or a novel sequence. ScanITD is commonly used to detect small to medium size duplications, but it does not perform well in samples with low coverage depth (Wang & Yang, 2020). SVIM uses graph-based clustering methods and a new signature distance metric to cluster detected SV signatures (Heller & Vingron, 2019). Although it achieves good results in all coverage ranges, SVIM does not perform well at low levels of tumor purity. TIDDIT uses discordant reads and split reads to detect the genomic location of SVs with RD signals, and it uses read depth information for variant classification and quality assessment (Eisfeldt et al., 2017). However, when the sequencing coverage is low, TIDDIT detection results have a higher false positive rate. Under the assumption of maximum parsimony, TARDIS integrates multiple sequence signatures to identify and cluster regions of potential SVs. It uses a probabilistic likelihood model to distinguish specific TDs by computing likelihood scores for each SV (Soylev et al., 2019). However, TARDIS is unsuitable for detecting sequencing data with low coverage and low tumor purity, and the detection results are relatively inaccurate.

Therefore, there are several challenges in detecting TDs: (1) The actual distribution of RD signals is uncertain due to the influence of noise from sequencing errors, mapping errors, and GC content bias in NGS, making the RD signal in the genome sequence insufficient for detecting variant regions; (2) When the coverage depth and tumor purity levels are low, sequencing data is particularly sensitive to noise, increasing the number of false-positive samples; (3) The distribution of sample categories is uneven, with variant regions usually accounting for a very small proportion of the genome. These challenges indicate that a more efficient method is needed to detect TDs.

This article proposes a TD detection method based on hybrid methods (DTDHM) that addresses these challenges. DTDHM uses NGS data from short-read sequencers to detect TDs in a single sample. DTDHM builds a pipeline that integrates RD, SR, and PEM signals. In the first stage of the pipeline, the k-nearest neighbor (KNN) algorithm is used to predict the candidate TD region. RD and mapping quality (MQ) signals are then integrated. These two signals are smoothed using the Total Variation (TV) model (Yuan et al., 2019), and the circular binary segmentation (CBS) algorithm (Venkatraman & Olshen, 2007) is used to segment the smoothed features. RD and MQ signals are then used as two features of the KNN algorithm (Zhang, 2022). The running result of the KNN algorithm declares outliers for each bin. Subsequently, a threshold value is set for outliers to reach the candidate TD region. In the second stage of the pipeline, split reads and discordant reads that conform to the definition of TDs are extracted to refine the boundary of TD regions. This article tests the performance of DTDHM on both simulated and real datasets and compares DTDHM with the existing methods of TD detection. The results show that DTDHM achieves a good balance of sensitivity and precision, significantly improving F1-score, maintaining an excellent overlap density score, and showing the best boundary accuracy for tandem duplication detection.

Materials and Methods

Workflow of DTDHM

DTDHM analyzes TDs without the need for control samples. The workflow of DTDHM is shown in Fig. 1 and consists of three main steps.

Figure 1 Workflow of the DTDHM.

DTDHM consists of three parts: data input, generation of information profile, and tandem duplication detection.

In the first step, RD and MQ are extracted from BAM files and preprocessed, accounting for missing values and the “N” positions in the reference genome. The sliding window splits the whole gene sequence into several continuous and non-overlapping bins. The RD and MQ are calculated for each bin and any bias caused by uneven GC content is corrected. The TV model is used to smooth the two signals and reduce noise. Then, the circular binary segmentation (CBS) algorithm is used to segment the bins into genome segments according to RD value fluctuations. Finally, RD and MQ are standardized.

In the second step, the candidate TD regions are obtained. RD and MQ are input into the DTDHM model as two features. The model calculates the outliers of each genome segment based on the KNN algorithm and detects some regions that may contain TDs as noise. The boxplot procedure sets an appropriate threshold for outliers. If the outlier in a region is greater than the threshold, it is considered a candidate TD region.

In the third step, the SR strategy is used to obtain split reads from BAM files. Then, CIGAR fields are used to infer the breakpoint position of the TDs. Finally, PEM is used to find and analyze discordant reads to refine the boundaries of TDs.

The DTDHM software is implemented primarily in Python. The following subsections describe each step of DTDHM in detail. 1) Dealing with missing values and “N” positions

The reference genome usually consists of four bases: “A,” “T,” “G,” and “C.” The character “N” is used to indicate that the position of the base has not been determined and could be any of the four bases mentioned above. There are missing values and “N” positions in any current version of the reference genome. These problems lead to mapping errors, where the read count deviates from the true read count. To provide a more reasonable read count, DTDHM fills the missing position with zero and adopts the removal strategy for the “N” position, as explained by Liu et al. (2020) and Yuan et al. (2019).

2) Obtaining RD and MQ

Due to the influence of noise from sequencing errors, mapping errors, and GC content bias, the RD signal does not follow a linear relationship with the TD region. Therefore, the variance of the RD signal is insufficient for detecting regions of variation in the genome sequence. MQ can indicate how reliably the read aligns to a position in the reference sequence and can be used in combination with other features for mutation detection (Zhao et al., 2020). For these reasons, DTDHM introduces MQ signals as one of the characteristics used for detecting TDs.

The genome sequence is divided into contiguous non-overlapping bins using a sliding window of fixed width. Because the “N” position in the reference sequence does not indicate whether tandem duplication events have occurred in the region, bins containing “N” are filtered. Then, the read pair count (RC) of each position in the bin is counted. The average value of RC in each bin is taken as the read depth signal of this bin, denoted as RD. The Formula (1) for calculating RD is as follows: (1) RDi=∑j=1Lb⁡RCijLb,iϵ[1,Nr],

where RDi represents the RD value of the i-th bin; RCij represents the RC value of the j-th position in the i-th bin; Lb represents the length of the genome bins, such as 1,000 bp; and Nr is the number of genomic bins generated from all regions.

MQ value is the average Mq value at each bin position. Its calculation method is similar to RD, and is calculated using the following Formula (2): (2) MQi=∑j=1Lb⁡MqijLb,iϵ[1,Nr],

where MQi indicates the MQ value of the i-th bin, and Mqij indicates the Mq value at the j-th position in the i-th bin.

3) Correction of GC content bias

NGS sequencing can lead to GC content bias during PCR amplification (Dohm et al., 2008). GC content bias affects the RD signal and then interferes with the prediction of tandem duplications. To eliminate the GC content bias, the GC fraction of each bin is calculated and then GC correction is performed on the RD signal of each bin (Yoon et al., 2009). The Formula (3) for GC correction is as follows: (3) RDi′=RD¯RD¯GC⋅RDi,iϵ[1,Nr],

where RDi′ and RDi represent the revised value and original value of the RD of the i-th bin, respectively; RD¯ is the mean of all RD’s; and RD¯GC represents the average RD that is similar to the GC score of the ith bin. Bins with similar GC scores are those bins with GC score differences less than or equal to 0.002.

4) Denoising using total variation

The RD and MQ between neighboring bins have a natural correlation. However, due to sequencing and mapping errors, the RD and MQ of adjacent bins may fluctuate randomly, resulting in noise. Therefore, a regular term is added to keep RD and MQ smooth. Unlike simple denoising methods, the TV model can remove noise while preserving the boundary information of the bins. Because of this, DTDHM uses the TV model to smooth and de-noise RD and MQ. The TV model is expressed as the solution to the minimization problem, and the Formula (4) is as follows: (4) minx∈RNr⁡{12‖y−x‖22+λ∑i=1Nr−1⁡|xi+1−xi|},

where x represents the signal containing noise; y represents the signal obtained after smoothing; both x and y are vectors, i.e., x=[x1,…,xNr]T and y=[y1,…,yNr]T; Nr represents the total number of bins; T is the matrix transpose; the first term represents the fitting error between x and y; the second term represents the total variation penalty; and λ represents the penalty parameter that controls the tradeoff between the first and second terms. Because choosing parameter λ can be a problem (Condat, 2013; Vaiter et al., 2013), DTDHM allows the user to specify the value of parameter λ, which is recommended to be selected from the range [0.15, 0.30]; the default is 0.15 (Yuan et al., 2021).

5) Region segmentation and standardization processing

Because the KNN algorithm is unsuitable for large sample sizes, DTDHM uses CBS to segment the bin-based genome into segments of different sizes, which reduces the computational complexity of KNN. RD and MQ are input into the KNN algorithm as two features. However, MQ and RD have a data imbalance problem because the value of MQ is much larger than the value of RD. When the KNN algorithm finds the nearest neighbor samples in the feature space, the contribution of MQ is greater than that of RD in the algorithm. To balance the influence of the two factors in the KNN algorithm, RD and MQ are normalized, as shown in the following Formula (5): (5) r′=(r−μ)/σ,

where r indicates RD or MQ, r′ is the standardized sample, μ represents the mean value of all sample data, and σ represents the standard deviation of all sample data.

Calling KNN to get candidate TDs

After preprocessing, the normalized RD and MQ are input into the KNN algorithm as two features. RD and MQ reflect magnitude of variation and mapping quality, respectively. The RD signal in one-dimensional space is transformed into a two-dimensional profile, D, combining RD and MQ (Yuan et al., 2019), as shown in Formula (6). The variation interval is analyzed from two perspectives to improve detection accuracy further. (6) D={(RDi,MQi)|,iϵ[1,Nr]}.

DTDHM adopts a KNN algorithm based on anomaly detection, which uses the basic principles of KNN but calculates outlier scores rather than predicting labels or values. The KNN anomaly detection algorithm is a distance-based method. Its core idea is to compute the distances between a genomic fragment point on a two-dimensional profile and its nearest k neighboring fragment points, evaluate the degree of anomaly of the sample through the size of the distance between the sample points, and judge whether it is a TD sample point. The characteristics of TDs cause the RD value and MQ value of the variant region to generally deviate from the value range of the normal region, and the k-nearest neighbor distance of a TD is usually much larger than that of a normal point. When using this rule to further calculate the outlier scores of genomic fragments, the larger the outlier scores of a genomic fragment, the more likely it is a TD. In this article, a genomic fragment is regarded as an object p.

In the KNN algorithm, in order to reduce the computational cost and speed up the search of k-nearest neighbors, DTDHM uses k-dimensional tree (KDTree), a high-dimensional index tree data structure. Using KDTree search, the k-nearest neighbor of any object in data space has a computational cost of O(Nlg⁡N). For the calculation of distance, DTDHM uses Euclidean distance. There are three common outlier scoring schemes: maximum, median, and average. The maximum and median outlier scoring schemes ignore the information from other nearest neighbors and exhibit instability. Because of this, DTDHM uses the more stable average outlier scoring scheme, which is the average distance between object p and all objects within its neighborhood.

When identifying tandem duplications, adjacent regions in the genome are positionally correlated (Yuan et al., 2017). After executing KNN and KDTree algorithms, the returned noise points can be regarded as a set of candidate TDs containing adjacent regions. Successive fragments are then merged to obtain candidate regions. However, because the KNN algorithm is sensitive to outliers, it can produce many false positives in the detection process, so the candidate regions then need to be filtered.

To accurately declare TDs based on outliers, a reasonable statistical model is needed to provide a cutoff point for outliers. The classical method produces an uneven distribution of sequencing reads, affecting the detection of low-amplitude TDs, so DTDHM uses the boxplot procedure to mark outliers (Sim & Chang, 2005) without any restrictions on the data (such as data following a certain distribution). This method uses interquartile range (IQR) to detect outliers. The interquartile range is the difference between the upper quartile (UQ) and the lower quartile (LQ) and contains half of all the data. Using parameter θ, the upper and lower values are defined as UQ+θ⋅IQR and LQ−θ⋅IQR. Objects that exceed the upper or lower limit of the boxplot are considered outliers. In tandem duplications, a higher outlier on the boxplot indicates a higher probability of an outlier, so the objects above the upper limit of the boxplot are considered candidate regions.

For an object p, Table 1 describes the steps KNN takes to declare a candidate TD. The KNN algorithm takes RD and MQ as features. It selects any sample point in the feature space and searches for its k-nearest neighbor domains using KDTree. Then, the average outlier score scheme is used to calculate the average distance between the sample point and its k-nearest neighbors, or the outlier score of the sample point. These outlier scores are obtained for all sample points, then a statistical analysis is carried out using the boxplot procedure. When the sample points fall above the upper limit of the boxplot, they are classified as outliers. The candidate TDs are obtained by counting all outliers.

Table 1 The steps KNN takes to declare a candidate TD.

Algorithm 1. The steps KNN takes to declare a candidate TD.	
1: For a given object p, k-nearest neighbors are searched by KDTree, and the nearest neighbor domain of the object p is obtained;	
2: Euclidean distance is used to calculate the distance between the object p and k-nearest neighbors;	
3: The average distance between object p and all objects in its nearest neighbor domain is calculated as the outlier score of object p;	
4: The threshold is set using the boxplot, and if the outlier score of the object p is higher than the threshold, it is marked as a potential TD.	

Filtering candidate regions

Candidate TD regions then have to be filtered for other types of structural variations, such as interspersed duplication. The SR strategy is used to identify and filter out the split reads that do not belong to the target tandem duplication. The SR strategy uses the local alignment algorithm to align the pair-end read pair with the reference genome and split the read segments that do not completely match the reference genome. Tandem and interspersed duplication can be distinguished by analyzing the characteristics and alignment of split reads. The CIGAR field in the BAM file indicates the alignment of the read segment with the reference genome, usually consisting of the characters “H,” “M,” and “S.” Split reads are determined by whether the CIGAR field contains an “S” cut symbol. In this article, CIGAR mainly analyzes two types of pre-alignment and post-alignment reads, as represented by Formula (7):

(7) CIGAR={xMySySxMx,y∈N,x+y=Rl,

where “M” means that the digit length in front of the symbol matches the reference position, and “S” means that it does not match and is soft cut. “xMyS” indicates pre-alignment; “ySxM” indicates post-alignment; and Rl represents the length of the short sequencing read segment, which is generally set to 100 bp. For example, if the length of the pair-end reads segment is 100 bp, the CIGAR field in one of the reads is “20S80M,” indicating that the read is post-alignment, the first 20 bp of the read does not match the reference position, but the remaining 80 bp does match the reference position.

Split reads in the alignment results are then scanned to find the set of potential breakpoints and summarize the SR characteristics of tandem and interspersed duplications. Figure 2 shows the SR characteristics of tandem and interspersed duplications. In Fig. 2A, R1 and R2 are two reads that span the junction of the repeat sequence. When the reads are mapped to the reference genome, both R1 and R2 are multiply mapped to points A and B. Point A represents the starting position of the variation, while point B represents the ending position of the variation. R1 and R2 are post-alignment (ySxM) at point A and pre-alignment (xMyS) at point B. This indicates that split reads generated by tandem duplications can map multiply to two breakpoints, which are, respectively, the starting position and the ending position of the tandem duplication. Furthermore, the alignment pattern at the breakpoints is consistent, being post-alignment at the starting position and pre-alignment at the ending position. In Fig. 2B, R3 and R4 are two split reads generated by interspersed duplication. R3 is multiply mapped to points A and C, while R4 is mapped to points B and C. Point C represents the insertion site of the interspersed duplication, with two alignment patterns (xMyS + ySxM). This shows that split reads generated by interspersed duplication can map to multiple breakpoints, with all breakpoints other than the starting and ending position including both pre-alignment and post-alignment.

Figure 2 The SR characteristics of tandem and interspersed duplications.

The orange area represents tandem duplications; R1, R2, R3, and R4 represent split reads and contain breakpoint information, such as “ySxM” and “xMyS” (see Formula (7) for their detailed definitions); Points A and B are the breakpoint locations detected according to SR; Point C represents the insertion site of the interspersed duplication. (A) An example of tandem duplication. (B) An example of interspersed duplication.

DTDHM uses the features of split reads to extract tandem duplications and filter out interspersed duplications from candidate regions. Post-alignment split reads are filtered out at the starting position of the candidate region, then other mapping positions of these reads are identified. If all split reads mapped to these other positions are pre-alignment and are located at the ending position of the candidate region, the candidate region is labeled as a tandem duplication; otherwise, it may be an interspersed duplication. Pre-alignment split reads are then filtered out at the ending position of the candidate region and the breakpoints of these reads are recorded. If all reads at the breakpoint position of this read are pre-alignment, and all reads at the other mapping positions of this read are post-alignment, the candidate region is labeled as a tandem duplication.

Refining boundaries

Because the KNN algorithm only uses RD and MQ signals, its detected candidate TD region boundary is inaccurate. To improve TD boundary accuracy to the bp level rather than the bin level, DTDHM optimizes the boundaries of TD regions based on the breakpoint positions of tandem duplications. There are many split reads in whole sequencing data and most split reads are useless for boundary detection, so DTDHM searches for split reads within the valid range. Figure 3 depicts an example of DTDHM exploring the effective split reads in a candidate TD region [a, b]. The length of m bins is used as the effective search step (S = m∙ Lb). Prior experience has indicated that most of the effective split reads gather in the range of two bins, and expanding that range may lead to noisy data, so m was set to 2. The starting position a and ending position b of the TD region detected are used as the search center. The detection range at both ends of the candidate TD region is 2S. Within this range, SRs containing a large amount of breakpoint information are extracted to correct the boundary of the candidate TD region [a, b]. Ideally, the boundary [a, b] can be precisely [A, B]. Using the SR strategy, the boundary of the TD region can be expressed by the following Formula (8):

Figure 3 An example of exploring the effective SR based on the detected TD region [a, b].

The orange region represents the candidate TD region with a rough boundary; Point a is the starting position of the rough TD region, and point b is the ending position; Point a and point b are the search center of SR exploration, and the search depth is S; The red dashed box represents SR’s effective search range (2S), which constitutes the extended TD region; R1 and R2 are the valid SR found in the extended region (2S); Point A and point B are the exact boundary positions of the candidate TD region.

(8) A=POS(Rpost(i)),B=POS(Rpre(j))+xRpre(j)−1,

where A represents the exact starting position of the candidate TD region; B represents the exact ending position of the candidate TD region; Rpost(i) represents a post-aligned split read in the range [a – S, a + S]; Rpre(j) represents a pre-aligned split read in the range [b – S, b + S]; POS() indicates the position where the read aligns with the reference genome; and xRpre(j) represents the length of the Rpre(j) that matches the reference genome. If R1 is post-alignment near position a, the exact starting position of the TD region can be determined by A=POS(Rpost(1)). If R2 is pre-alignment near position b, the exact ending position of the TD region can be determined by B=POS(Rpre(2))+xRpre(2)−1. Finally, the rough boundary [a, b] of the TD region is exactly [A, B].

However, SR can only refine a limited area at low coverage depth and tumor purity. After processing with SR, some boundaries of TD regions [a, b] remain rough, such as [a, B], [A, b], and [a, b]. Therefore, the PEM strategy is then used to further deal with the boundaries not covered by SR. Figure 4 shows an example of PEM in tandem duplication. This strategy focuses on pair-end reads spanning tandem duplication junctions that align to the reference sequence to form discordant read pairs. Among these discordant read pairs, the read (R2′) at the front of the alignment position is closest to the starting position of the TD region. Similarly, the read (R5′) at the back of the alignment position is closest to the ending position of the TD region. Using this rule, a more accurate TD region is obtained.

Figure 4 An example of PEM in tandem duplication.

The orange area represents tandem duplications; Points A and B are TD breakpoints; R1 and R2, R3 and R4, and R5 and R6 are three pair-end reads spanning the tandem duplication junction; R1′ and R2′, R3′ and R4′, R5′ and R6′ are three discordant read pairs.

Figure 5 depicts an example of using PEM signals to refine candidate TD boundaries. Since SR signals have been previously used to refine the boundaries of TDs, only the reads without SR signals are included when processing PEM signals. First, the location of the extracted discordant read pair must be within the search radius S of the starting position (which may contain the true breakpoint information). Second, the absolute value of the insert size of discordant reads must be within a certain range of the length of the detected candidate TD region (the default range is 2S). DTDHM sorts discordant reads (such as R4′ and R6′) that meet these two conditions by the size of the alignment position. The alignment position of read R4′ is the most forward, and the alignment position of read R6′ is the most backward. The position of R5′, another pair-end read of R6′, is closest to the TD ending position. According to these rules, the alignment position of read R4′ can approximate the TD starting position, and the position of R5′ plus the length of the read segment ( Rl) can approximate the TD ending position. Using the pair-end mapping strategy, the boundary of the TD region can be expressed by the following Formula (9):

Figure 5 An example of using PEM signals to refine candidate TD boundaries.

The orange region represents the candidate TD region with a rough boundary; Point a and point b are the starting and ending positions of the candidate TD region; Point A and point B are the exact boundary positions of the candidate TD region, respectively; The red dashed box represents PEM’s effective search range (2S); R3′ and R4′, R5′ and R6′ are two discordant read pairs, where R4′ and R6′ are within the effective search range (2S).

(9) A=POS(Rfirst(i)),B=POS(PnextRlast(j))+Rl,

where Rfirst(i) represents the read pair with the smallest comparison position among the discordant read pairs that meet two conditions; Rlast(j) represents the read pair with the largest comparison position among the discordant read pairs that meet all conditions; and PnextRlast(j) represents the other read pair corresponding to Rlast(j) in paired-end sequencing; Rl is defined in Formula (7).

Table 2 details the process of declaring TDs and the precise location of TD boundaries.

Table 2 The process of declaring TDs.

Algorithm 2. The process of declaring TDs.	
1: RD and MQ are entered into the KNN algorithm and the outlier scores Sp are calculated for each genome fragment;	
2: Set the threshold θ, and if the outlier scores Sp>θ, the genome segment is considered a candidate region, with boundaries [a, b];	
3: Using SR strategy to select tandem duplications from candidate TDs regions;	
4: The split reads are explored at the [a − S, a + S] and [b – S, b + S] ranges at both ends of the candidate TD region. The candidate TD region is modified according to Formula (8);	
5: If starting position = a or ending position = b, the discordant read pairs that meet all conditions are sought in the boundary [starting position – S, starting position + S] of the candidate TD. Based on the positional information of these discordant read pairs, modify the TD boundary according to Formula (9);	
6: Repeat steps 4 and 5 until all TD regions are scanned.	

Results and discussion

The DTDHM software is implemented in Python language based on the methods described above, and the code is publicly available at https://GitHub.com/yuantt75/DTDHM.git.

Both simulated and real datasets were used to evaluate and verify the performance of DTDHM. In simulated data experiments, DTDHM was compared with SVIM, TARDIS, and TIDDIT in four aspects of performance: sensitivity, precision, F1-score (average harmonic value of sensitivity and precision), and boundary bias. Five real sequencing samples from the 1,000 Genome Project (http://www.1000genomes.org; Genomes Project C et al., 2015) were used to test DTDHM on real data. F1-score and overlap density score (ODS) were used to evaluate the performance of each method. In addition, the run time and memory consumption of these methods were analyzed and the comparison results are shown in the Supplemental File. During the experiment, the length of genomic bins ( Lb) was 1,000 bp. The sample genome length was divided by Lb and rounded upward to obtain the number of genome bins as Nr. The value of the default integer k in the KNN algorithm was 20% of the sample size (0.2 Nr). The default value of θ in the boxplot was 0.6. Detailed parameter selections can be found in the Supplemental File.

Simulation studies

In generating simulated data, Python was used to generate a FASTA file containing variation information. SinC (Pattnaik et al., 2013) and seqtk (https://GitHub.com/lh3/seqtk) software were combined to generate datasets of different levels of tumor purity and coverage depth. The SInC_readGen program in the SInC software can generate two types of FASTQ files based on FASTA files: one is normal and the other contains mutation information. The “−R” and “−C” parameters in the SInC_readGen program can specify read length and coverage depth. “−R” was set to 100. It is worth noting that due to the paired-end sequencing used in this study, merging a pair of read FASTQ files doubled the coverage depth of the merged FASTQ file; therefore, the value of “−C” should be half of the actual coverage depth. Next, the sub-sampling feature in Seqtk was used to extract data from both the normal and mutant FASTQ files to generate data with different levels of tumor purity. Then, the sequencing reads were mapped to the reference genome using the BWA-MEM (Li & Durbin, 2010) tool to generate a SAM file. Finally, SAMtools (Li et al., 2009) software was used to convert the files from SAM format to BAM format and sort the BAM files.

In the simulated dataset, tumor purity was set to 0.2, 0.4, and 0.6, and coverage depth was set to 4, 6, and 8X. In each simulation configuration, the size of tandem and interspersed duplications ranged from 10 to 50 kbp. Chromosome 21 in GRCh38 was selected as the reference genome. Each configuration generated 50 samples and their average sensitivity, precision, and F1-score were calculated to reduce the randomness of the experiment. DTDHM and three existing methods were tested on the simulated dataset. A true positive was recorded when the detection result covered more than half of a true TD area. The calculation methods for sensitivity, precision, and F1-score are shown in the following Formulas (10)–(12): (10) Sensitivity=TPP

(11) Precision=TPTP+FP

(12) F1−score=2×Precision×SensitivityPrecision+Sensitivity

where sensitivity, also known as recall, refers to the ratio of correctly predicted tandem duplication count (TP) to the total actual tandem duplication count (P); FP is the count of tandem duplications incorrectly predicted by the method; precision refers to the ratio of correctly predicted tandem duplication count (TP) to the total predicted tandem duplication count (TP + FP); and F1-score represents the harmonic mean of sensitivity and precision, two mutually balancing metrics. A higher F1-score indicates better performance of the detection method.

Table 3 shows the F1-score of the four methods in different configurations. Among all coverage depth and tumor purity, DTDHM consistently maintained the highest F1-score. In 450 simulated data samples, the average F1-score of DTDHM, SVIM, TARDIS, and TIDDIT were 80.0%, 56.2%, 43.4%, and 67.1%, respectively. The F1-score of DTDHM was 1.2 times higher than that of the second-best method.

Table 3 F1-score of the four methods in different configurations.

Sample	DTDHM	SVIM	TARDIS	TIDDIT	
4X_0.2	0.695	0.435	0.060	0.428	
4X_0.4	0.779	0.547	0.344	0.670	
4X_0.6	0.859	0.606	0.495	0.743	
6X_0.2	0.687	0.491	0.227	0.559	
6X_0.4	0.771	0.587	0.508	0.711	
6X_0.6	0.834	0.617	0.625	0.771	
8X_0.2	0.782	0.550	0.350	0.623	
8X_0.4	0.870	0.605	0.605	0.727	
8X_0.6	0.920	0.619	0.692	0.807	
Average	0.800	0.562	0.434	0.671	
Note:

The bold indicates the optimal value.

Figure 6 depicts the F1-score comparison of the four algorithms in different configurations. The F1-score of these four methods improved significantly as coverage depth and tumor purity increased. The DTDHM method exhibited a relatively small variation in F1-score, indicating stable detection performance. At the coverage depth of 4X, with the change of tumor purity, the F1-score of DTDHM changed from 85.9% to 69.5%, increasing by 16.4%, while the F1-score of SVIM increased by 17.2%, the F1-score of TARDIS increased by 43.5%, and the F1-score of TIDDIT increased by 31.5%. At 6X and 8X coverage depths, the F1-scores of DTDHM improved by 14.7% and 13.8%, respectively. TARDIS was the most sensitive to changes in tumor purity, followed by TIDDIT. All methods were most effective when the tumor purity was 0.6. The results of the four methods were more stable at different coverage depths than they were with changes to tumor purity.

Figure 6 The comparison of F1-score of the four algorithms in different configurations.

The horizontal axis represents coverage depth and tumor purity, while the vertical axis represents F1-score.

Figure 7 compares the sensitivity, precision, and F1-score between DTDHM and the three methods on simulated datasets. Under each set of conditions, DTDHM achieved the best F1-score, followed by TIDDIT, SVIM, and TARDIS. The average sensitivity of DTDHM was the highest at 79.9%, which was 21.9%, 47.7%, and 13.5% higher than SVIM, TARDIS, and TIDDIT, respectively. The average precision of DTDHM was 80.7%, and the average precision of SVIM, TARDIS, and TIDDIT were 57%, 82.7%, and 73.2%, respectively. SVIM achieved the lowest precision under each set of conditions, meaning it detected many TDs, most of which were false positives. TARDIS achieved the lowest sensitivity but had a higher precision, indicating this method was more conservative in its detection than the other methods. The performance of TIDDIT was better than that of both SVIM and TARDIS, ranking second in terms of F1-score. Of all the tested methods, DTDHM achieved the best balance of precision and sensitivity.

Figure 7 Comparison of sensitivity, precision, and F1-score between DTDHM and the three methods on simulated datasets.

The F1-score is shown as a black curve, ranging from 0.1 to 0.9. “Cov” is short for coverage depth.

For the detection accuracy of TD boundaries, the boundary bias of 50 sets of experiments under each configuration were plotted to better show the distribution of data. The boundary bias was defined as the average base pair (bp) of the recalled TDs from the actual TD boundary deviation. The smaller the boundary bias, the more accurate the TD detection method. The comparison of boundary bias of the four methods is shown in Fig. 8. As seen in Fig. 8, the boundary bias of the four methods decreased as tumor purity and coverage depth increased. This is due to the fact that there are fewer effective SR and PEM signals in low tumor purity samples, leading to inaccurate breakpoint identification. Overall, the SVIM method exhibited smaller boundary biases at 4X coverage depth with boundary biases consistently around 36 bp, followed by the DTDHM method, with most boundary biases around 50 bp. For 6X and 8X coverage depth, DTDHM demonstrated smaller boundary biases with an average bias of 34 bp, followed by SVIM. The TARDIS method had the worst and most volatile results. The detection of these methods was also analyzed in the 10X_0.6 configuration, and the results are presented in the Supplemental File. The results of these comparisons of simulated data indicate that DTDHM is an effective and reliable TD detection method.

Figure 8 The boundary bias of DTDHM and three other methods is compared on nine simulated datasets.

Application to the real samples

Because tests on simulated data cannot reflect the real detection effect of the algorithm on biological data, DTDHM was also used to analyze five real sequencing samples (NA19238, NA19239, NA19240, HG00266, and NA12891) from the 1,000 Genomes Project. The first three samples were from the YRI trio of the Yoruba family, the fourth sample was a woman of European descent from Finland, and the fifth sample was from the CEU trio of the European ancestry. TDs were predicted from a single tumor sample using whole genome sequencing (WGS) data.

DTDHM was compared with three other methods (SVIM, TARDIS, and TIDDIT). Due to the lack of ground truth files for the first four samples, it was not possible to calculate sensitivity, precision, and F1-score. Therefore, the detection effects of the four methods were analyzed by calculating the ODS of each method (Liu et al., 2020; Yuan et al., 2020, 2021). The calculation of ODS is shown in Formula (13):

(13) ODS=AverOlap⋅RatioOlap,

where AverOlap represents the average number of overlapping events between an algorithm and other algorithms, or the total overlapping TDs divided by the number of compared algorithms, which is roughly the number of overlapping TDs detected by the algorithm; and RatioOlap represents the ratio of the average number of overlapping events of the algorithm to the number of predicted events. Overlapping events refer to pairwise intersections of prediction results between any algorithms. In this article, overlapping events between different algorithms were assumed to be true positives, so RatioOlap was considered an indicator of precision.

Figure 9 shows the number of overlapping TDs predicted by the algorithm on each chromosome, in kbp. In the chord diagram, the upper half of the circle is divided into four sections denoting DTDHM, SVIM, TARDIS, and TIDDIT. The lower half of the circle is divided into 22 sections, representing autosomes 1 through 22. As shown in Fig. 9, TIDDIT detected the highest total number of overlapping TDs, followed by DTDHM, TARDIS, and SVIM. Tandem duplication structures are usually different for different chromosomes. DTDHM detected the most overlapping TDs on chromosome 1, followed by chromosome 10. On chromosome 10, the total number of overlapping events between DTDHM and other algorithms was 141.7kbp, while the total number of overlapping events of SVIM, TARDIS, and TIDDIT compared to the other algorithms were 84.36, 104.73, and 89.84 kbp, respectively. These results indicate that the four algorithms achieved good prediction results on chromosome 10.

Figure 9 Overview of the distribution of overlapping TD events detected by the four methods in four real samples (NA19238, NA19239, NA19240, and HG00266) in kbp.

In the upper half of the circle, the purple, green, blue, and red arcs represent DTDHM, SVIM, TARDIS, and TIDDIT, respectively. The grey arcs in the lower half of the circle represent autosomes.

Figure 10 records the average number of overlapping events ( AverOlap) per chromosome in four samples for the four methods. Table 4 shows the ODS calculation results for these four methods. As shown in Fig. 10, TIDDIT detected the most number of overlapping TDs in all four samples due to the fact that TIDDIT predicts the largest number of total events. However, the precision of TIDDIT was not high. In sample NA19240, TIDDIT detected 15,059 bp overlapping TDs, but its ODS was only 945.38, which is much lower than DTDHM’s ODS of 2,503.12. This indicates that detecting a large number of TDs may cause too many false positives. TARDIS detected fewer TDs, but its ODS was lower than that of DTDHM. DTDHM detected a modest number of TDs in these samples and had a relatively higher overlapping density than the other methods.

Figure 10 The average number of overlapping events per chromosome in four real samples for the four methods.

Table 4 The ODS of four methods in four samples of the whole genome (22 autosomal).

Sample	DTDHM	SVIM	TARDIS	TIDDIT	
NA19238	2,800.10	125.24	2,754.42	1,435.24	
NA19239	2,358.74	743.48	2,596.60	1,729.40	
NA19240	2,503.12	234.10	2,223.79	945.38	
HG00266	2,200.52	1,183.38	1,525.85	741.62	
Note:

The bold indicates the optimal value.

Table 4 shows that DTDHM achieved the highest ODS in three of the real samples, followed by TARDIS, TIDDIT, and SVIM. In the four real samples, the ODS of DTDHM was 1.02, 0.94, 1.13, and 1.44 times the ODS of TARDIS, respectively. TARDIS had a relatively high ODS in each real sample, but its low AverOlap resulted in poorer detection performance compared to DTDHM. DTDHM maintained a high AverOlap and RatioOlap and had the best ODS among the real samples.

The long-read sequencing data results of chromosome 21 for NA12891 was downloaded from the NCBI database to provide a more intuitive display of method performance (https://www.ncbi.nlm.nih.gov/). Each method was repeated 20 times for each sample, and the average performance metrics of precision, sensitivity, and F1-score were recorded for each method. As shown in Fig. 11, DTDHM demonstrated the best performance on NA12891, achieving the highest F1-score. DTDHM was able to balance precision and sensitivity to provide relatively accurate results. On NA12891, DTDHM’s F1-score was 42.1%, which was 17.1%, 20.7%, and 22.1% higher than SVIM, TARDIS, and TIDDIT, respectively. The results for TARDIS and TIDDIT were very similar. Furthermore, the precision and sensitivity of DTDHM were superior to other methods. It is worth noting that, in contrast to the results from the simulated datasets (Fig. 7), the results of many methods were not satisfactory on chromosome 21 for NA12891. This is because the types of variations in the real genome are more complex, leading to deviations in method performance between the simulated dataset and the real dataset. These results indicate that the application of the DTDHM method in actual data is relatively reliable.

Figure 11 The performance of four methods is evaluated on NA12891.

The F1-score is shown as a black curve, ranging from 0.1 to 0.9. “Cov” is short for coverage depth.

Discussion

This article proposes a new method of tandem duplication detection called DTDHM. This method uses NGS data from short-read sequencers to detect TDs in a single sample. DTDHM uses a classification method in machine learning and combines RD, SR, and PEM strategies. In the first phase, it uses RD and MQ signals to build a two-dimensional array to detect the TD regions. In the second phase, it uses SR and PEM strategies to screen and refine TD regions.

The DTDHM method uses the individual strengths of RD, SR, and PEM strategies. The RD strategy is good at detecting duplications and deletions. When the RD signal of a certain segment is significantly higher than the average RD signal, the segment may be repeated. DTDHM uses RD signals to screen out potential regions of tandem repeat variations from the data, greatly improving the F1-score and overlap density score of the detection results. DTDHM uses the SR strategy to extract tandem duplication features from the CIGAR field of split read alignments, enabling it to both identify TD regions from potential duplications and precisely determine the boundaries of detection results. The SR strategy plays a significant role in improving detection precision. However, the SR strategy is not capable of refining the boundaries of some TD regions because of its low coverage depth and low tumor purity, so DTDHM uses the PEM strategy to further process the boundaries that are not refined by the SR strategy. The RD strategy contributes the most to F1-score and overlap density score, followed by the SR strategy. The SR strategy contributes the most to boundary accuracy, followed by the PEM strategy.

Three feature benefits of DTDHM can be summarized as follows: (1) DTDHM can effectively solve the problem of an unbalanced distribution of normal and abnormal samples without assuming the data distribution. DTDHM adopts nonlinear classification. By integrating RD and MQ signals, the anomaly of each region is effectively characterized, which can reduce the influence of noise caused by mapping and sequencing errors. (2) DTDHM assigns an outlier score to each sample according to its first k nearest neighbor distances. This method can be easily extended to other data types and improves DTDHM’s ability to detect local TDs. (3) In the case of low coverage depth and tumor purity, DTDHM has better precision and sensitivity than most methods.

Conclusions

This article introduces the DTDHM method of tandem duplication detection and then tests it on 450 simulated datasets and five real datasets. The simulation experiment compared DTDHM with three other methods under different data configurations. The results show that DTDHM achieved the best balance of sensitivity, precision, F1-score, and boundary bias. In real data applications, DTDHM performed better than the other methods in terms of ODS and F1-score. These results indicate that DTDHM is a reliable tool for detecting TDs from NGS data, especially in the case of low coverage depth and low tumor purity samples.

However, DTDHM currently has limitations. DTDHM can only detect TDs, and the detection of other mutations, such as CNV or interspersed duplication, should be added in future work. Secondly, DTDHM uses the KNN algorithm to detect TDs, and other anomaly detection algorithms should be studied in the future to obtain better performance. Future studies are planned to improve DTDHM.

Supplemental Information

Supplemental Information 1 Supplementa. Figure and Tables.

Additional Information and Declarations

Competing Interests

Author Contributions

Data Availability

The authors declare that they have no competing interests.

Tianting Yuan conceived and designed the experiments, prepared figures and/or tables, and approved the final draft.

Jinxin Dong conceived and designed the experiments, authored or reviewed drafts of the article, and approved the final draft.

Baoxian Jia analyzed the data, authored or reviewed drafts of the article, and approved the final draft.

Hua Jiang analyzed the data, authored or reviewed drafts of the article, and approved the final draft.

Zuyao Zhao performed the experiments, prepared figures and/or tables, and approved the final draft.

Mengjiao Zhou performed the experiments, prepared figures and/or tables, and approved the final draft.

The following information was supplied regarding data availability:

The DTDHM: Detection of Tandem Duplications Based on Hybrid Methods Using Next-Generation Sequencing Data is available at GitHub and Zenodo:

https://GitHub.com/yuantt75/DTDHM.

- yuantt. (2024). yuantt75/DTDHM: DTDHM code (v1.0.0). Zenodo.

https://doi.org/10.5281/zenodo.12619197.

Yuan, Tianting (2024). Simulated dataset for the DTDHM method. figshare. Dataset.

https://doi.org/10.6084/m9.figshare.25001882.v1.

The genomes are available at the International Genome Sample Resource: Sample NA19238, Sample NA19239, Sample NA19240, Sample HG00266, Sample NA12891.

https://www.internationalgenome.org/data-portal/sample/NA19238

https://www.internationalgenome.org/data-portal/sample/NA19239

https://www.internationalgenome.org/data-portal/sample/NA19240

https://www.internationalgenome.org/data-portal/sample/HG00266

https://www.internationalgenome.org/data-portal/sample/NA12891

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
