# Peer review of "DTDHM: detection of tandem duplications based on hybrid methods using next-generation sequencing data"

_PeerJ, doi:10.7717/peerj.17748_

## Round 0.1 · original submission · Major Revisions

Thank you for submitting your manuscript, which presents a novel method, DTDHM, for identifying tandem duplicates and opens an intriguing avenue for the application of hybrid methods in the analysis of Next-Generation Sequencing (NGS) data. Your work certainly casts a positive light on the potential of hybrid approaches and has the potential to draw significant attention within the scientific community.

However, upon reviewing feedback from two peer reviewers, there are several concerns that need to be addressed to strengthen your manuscript. The reviewers have pointed out critical issues regarding the research design, notably the absence of clear guidance on the selection of parameters and questions regarding the reproducibility of your findings. Such issues are pivotal for the scientific community to validate and build upon your work.

Additionally, one review highlighted a significant oversight: the manuscript does not consider other types of structural variations that could also be detected by DTDHM. It is essential to test, explain, and justify the omission of these variations to ensure a comprehensive understanding of DTDHM's capabilities and limitations.

The reviewers also provided constructive suggestions to enhance the paper's readability and overall flow. Improving these aspects will not only make your paper more accessible but also ensure that your message is clearly communicated.

In light of the feedback provided, I strongly recommend addressing the highlighted issues to the fullest extent possible before submitting your revised manuscript. Incorporating these changes will significantly improve the quality and impact of your research. We look forward to receiving your revised submission and believe that your work has the potential to make a meaningful contribution to the field.

**Language Note:** The review process has identified that the English language must be improved. PeerJ can provide language editing services - please contact us at [email protected] for pricing (be sure to provide your manuscript number and title). Alternatively, you should make your own arrangements to improve the language quality and provide details in your response letter. – PeerJ Staff

Reviewer 1 ·

Basic reporting

The English language should be improved so that the definitions and method descriptions are clear. Some examples where the language could be improved include lines 177 (which bins are used to calculate the average values when determining RD_GC?), 198 (why the term ‘eigenvalue’ is used in relation to the input data in kNN?) and 439 (it is unclear whether any pairwise intersections of predictions are taken into account, or only those predicted by all four algorithms). Similarly, it is not clear which overlaps are meant on figures 9 and 10, and there are no overlaps between the arcs on figure 9.

The DTDHM source code is published on GitHub along with a description of the required input files, and I suggest adding the following information: (1) a description of the output format and (2) the versions of the dependencies used.

Experimental design

The paper relies on the synthetic dataset for the comparison of DTDHM method with the existing methods for structural variation detection. Please describe the exact protocol for rearranged genome simulation, i.e. which parameters are used for the SinC simulation? Could you try DTDHM on data generated by simulation tools based on different approaches than SinC, such as VarSim that inserts known variants into reference?

In the validation on the real data, the motivation for ODS score metric is unclear, e.g. how is the average number of overlapping events, Aver_Olap, similar to sensitivity?

The read mapping is the prerequisite for DTDHM analysis, and the parameters used on this step may greatly change the number of split reads and discordant read pairs. Please specify which mapping software and which parameters were used for simulation and real dataset read alignment.

In my opinion, you should change focus of your Materials and methods section from the detailed definitions for all the methods used (for example, kNN description in lines 214-269) towards the information required to reproduce your analysis. I propose to provide a detailed description of the novel procedures, while for known methods a short description with the values of key parameters and a reference to the program package, if used (for example, kNN method from the specific version of pyod package), is sufficient.

Validity of the findings

The paper shows that DTDHM overperforms the existing software for structural variation events detection (TARDIS, TIDDIT and SVIM) on simulated dataset and real short read data. For the short read data used, no ground truth information is available, so the intersection of different methods predictions is used as the true positive set in the paper. I propose using a dataset with orthogonal long-read sequencing data to use the detected tandem duplications as ground truth instead.

The DTDHM method is designed for tandem duplication detection, however, there are structural variation events of other types, such as interspersed duplications, which also result in higher read coverage. As I see it, they will be reported by DTDHM as candidate TD regions, but there will be no split reads supporting these candidate regions as shown on Figure 2. Please describe the handling of such events in your workflow and, if possible, test DTDHM on simulations with these events to see if it handles them correctly.

·

Basic reporting

Thank you for developing DTDHM. The presentation is clear and the manuscript is well written. The authors describe DTDHM, a tool to detect tandem duplications using an ensembled pipeline of paired-end mapping (PEM), split read (SR) and read depth (RD) approaches. It identifies the outliers using K-NN as candidate tandem duplications, then removes the false positives using boxplot statistics. DTDHM achieved the best F1 scores, overlap density scores, and boundary accuracies.

Experimental design

1) DTDHM is an ensemble pipeline, but it is not sure what part of the model contributes to what part of the results. Can you explain which of the PEM, SR, and RD contributed most to the F1 score, overlap density score, and boundary accuracy, respectively?

2) There are many important parameters during the process of DTDHM (lines 378-381). Please give justifications for how these parameters were selected.

3) The draft lacks an analysis of the time and memory consumption of the benchmarked tools.

4) Can you explain the rationale for using k-NN as an outlier detection tool? There are many other outlier detection methods that might work.

Validity of the findings

1) I cannot find the formula for the F1 score. How are the precision and recall for the F1-score calculated?

2) In Figure 6, F1 scores of DTDHM and TIDDIT are close for 8x_0.6. Can you show the results of 10x_0.6? I suspect DTDHM has a smaller advantage among the other tools for high depth.

3) Please explain the possible reasons why DTDHM is lower than TIDDIT in the number of overlapping events in Figure 10.

Additional comments

Line 221: “split” should be “splits”.

---

## Round 0.2 · accepted · Accept

All the concerns of the reviewers have been resolved, thank you for your patience and thorough revision of the article. Invited reviewers and I are very happy with the current version which has significantly improved since the first submission. The manuscript is now generally ready for publication, and I am pleased that I was able to interact with you in this journey.

Reviewer 1 ·

Basic reporting

The authors have addressed the comments made by me and the other reviewer. I am quite satisfied with the reviewed manuscript.

Experimental design

no comment

Validity of the findings

no comment

Additional comments

no comment

·

Basic reporting

The manuscript has been improved a lot after the revision. I agree the manuscript to be accepted by PeerJ.

Experimental design

No additional comments.

Validity of the findings

No additional comments.

Additional comments

No additional comments.